# Cellular and Molecular Basis of Environment-Induced Color Change in a Tree Frog

**DOI:** 10.3390/ani14233472

**Published:** 2024-12-01

**Authors:** Runliang Zhai, Liming Chang, Jianping Jiang, Bin Wang, Wei Zhu

**Affiliations:** 1Chengdu Institute of Biology, Chinese Academy of Sciences, Chengdu 610213, China; zhairunliang22@mails.ucas.ac.cn (R.Z.); changlm@cib.ac.cn (L.C.); jiangjp@cib.ac.cn (J.J.); 2University of Chinese Academy of Sciences, Beijing 101408, China

**Keywords:** scRNA-seq, amphibian, background adaptation, skin chromatophore

## Abstract

For a long time, scientists have been fascinated by how amphibians change color to blend into their surroundings. However, the exact cellular mechanisms behind this phenomenon remain unclear. In this study, we investigated the tree frog species *Rhacophorus dugritei*, which can change its skin color to adapt to different environments. Using single-cell RNA sequencing, we analyzed the frog’s skin cells and identified two distinct types of pigment cells responsible for color change. Our findings suggest that the number of these cells plays a more crucial role in color change than their individual activity. Additionally, we discovered that darker frogs exhibited higher levels of energy metabolism in their skin cells, while lighter-colored frogs showed increased immune-related activity. These findings indicate that amphibian color adaptation involves not only visible changes in skin pigmentation but also underlying biological processes. This research enhances our understanding of how animals adapt to their environments and may have important implications for studies on animal behavior, evolution, and conservation.

## 1. Introduction

The remarkable ability of vertebrates to undergo rapid color changes in response to environmental factors, particularly background color, is a captivating adaptive mechanism that enhances survival [1,2,3]. This phenomenon, observed in amphibians, reptiles, and certain fish, plays a critical role in facilitating camouflage [4,5], communication [6], and thermoregulation [7,8]. Understanding the cellular and molecular mechanisms underlying these color changes not only illuminates evolutionary adaptations but also holds significant value in various practical applications, including biomimicry and adaptive materials [9,10].

Chromatophores are pigment cells in ectothermic vertebrates that generate color by producing pigments or reflecting light [11]. They facilitate dynamic color changes, allowing adaptation to environmental conditions. Chromatophores are classified into six types: melanophores (black or brown melanin), xanthophores (yellow pigments from pteridines and carotenoids), erythrophores (red pigments from carotenoids) [12], iridophores and leucophores (guanine platelets for metallic or reflective white colors) [13], and cyanophores (blue pigmentation in certain species, such as fish) [14]. The interplay and regulation of these types, driven by specific pigment synthesis genes, underpin the complex color changes observed in ectothermic vertebrates [15]. For instance, genes involved in the synthesis of melanin, such as *TYRP1* and *TYR*, are crucial for the function of melanophores [16], while those associated with pteridine synthesis (e.g., *GCH1* and *PAX7*) are essential for xanthophores [17]. Color change in lower vertebrates is always contributed by synchronized changes in different types of chromatophores. There were two main types of color changes in vertebrates: morphological and physiological color changes (MCC or PCC) [15]. MCC primarily involves the differentiation, proliferation, and development of melanophores, leading to changes in the density and distribution of pigments within the skin [18]. This process is often regulated by genes associated with melanogenesis, such as *MITF*, *TYRP1*, and *DCT* [19], which play crucial roles in melanin production and melanophore function. In contrast, PCC occurs more rapidly and is driven by the transport, dispersion, and aggregation of existing melanin within chromatophores [20]. This process is accomplished through the precise coordination of molecular components, including the cytoskeleton, motor proteins, cargo proteins, and the cargo-recognition protein in melanophore (e.g., *MREG*) [21].

The background matching or adaptation observed in ectothermic vertebrates is considered to be primarily attributed to PCC [2,21,22,23,24]. However, in natural conditions, the matching of skin color to the background can occur over time scales ranging from minutes to hours, days, or even weeks [25,26]. This suggests a potential role for morphological color change (MCC) in skin color adaptation; however, the underlying cellular and molecular processes have been rarely studied. One key inquiry in this context is whether the observed color changes in response to background are primarily determined by pigment content, the number of pigment cells, or their distribution patterns [27]. It is also important to investigate whether different types of chromatophores respond differently to background color [28] and whether certain chromatophores play a more dominant role in this process [29]. Additionally, despite significant advances in our understanding of color change mechanisms, several critical questions remain unanswered [30]. For instance, do the physiological status and activities of pigment cells differ among individuals exhibiting different skin color conditions? Addressing these questions is essential for a comprehensive understanding of the factors driving color adaptation in response to environmental cues. Recent advancements in single-cell RNA sequencing (scRNA-seq) technologies provide exciting opportunities to study these mechanisms at an unprecedented resolution [31]. By examining individual cells, researchers can gain insights into the specific types of pigment cells involved in amphibian responses to environmental background color. This approach allows for a comprehensive understanding of the heterogeneity within pigment cell populations and the molecular pathways activated during pigmentation [32].

Tree frogs have been widely used as model organisms in studies of background adaptation, and this physiological response is considered crucial for their crypsis and thermoregulation [22,23,24,33,34]. The tree frog *Rhacophorus dugritei* (Anura: Rhacophoridae) exhibits a remarkable ability to adjust its skin color in response to changes in environmental backgrounds. With a high-quality genome reported, this species provides an excellent foundation for single-cell sequencing research, making it an ideal model for studying vertebrate responses to background colors. This study aims to investigate the cellular and molecular mechanisms underlying skin color changes in *R. dugritei* in response to black-and-white backgrounds. The primary scientific questions for this study were as follows: (1) Is the environmental background matching in this species primarily a morphological or physiological color change? (2) What types of chromatophores were involved in amphibians’ responses to environmental background colors? (3) What molecular differences exist in chromatophores under varying color states? We anticipate that the answers to these questions will not only enhance our understanding of amphibian biology but also contribute to broader ecological and evolutionary discussions about how organisms adapt to their environments.

## 2. Materials and Methods

### 2.1. Animal Collection and Acclimation

Thirty adult male *R. dugritei* were collected from Wawu Mountain, Huanya County, Meishan City (104.084536° E, 30.635704° N) in July 2022. Only male individuals were included in this study to minimize the effects of sex differences on hormone levels and gene expression. Additionally, as this species had a low proportion of females in the wild, the collection of female individuals could have a significant impact on the population. Before the experiment, each individual was placed in a separate transparent white plastic box (20 cm × 15 cm × 12 cm) and fed cherry roaches once daily. These animals were acclimated to an artificial climate incubator (YANGHUI Instrument Co., Ltd., Ningbo, China) for seven days at 20 ± 1 °C with a 12 h light/12 h dark cycle. Illumination was provided by the incubator’s built-in LED lamps, set to 20% of the maximum output (equivalent to approximately 4000 Lx, based on a range of 0–20000 Lx).

During the experiment, the frogs were transferred into nontransparent plastic boxes (25 cm × 18 cm × 15 cm), which were black or white. The lid of the boxes had a bait grid that allowed light to enter. The frogs were randomly divided into black and white boxes, with the other conditions kept the same. The morphology and skin color of the frogs were recorded with digital cameras at 36 (light phase), 48 (dark phase), 60 (light phase), and 72 (dark phase) hours after acclimation. After 72 h of treatment, the eight frogs were euthanatized using 0.2% MS-222 and collected the dorsal skin tissues (approximately 1 × 1 cm^2^ per sample). These skin tissues were cut into pieces in tissue storage solution (Miltenyi Biotec, Bergisch Gladbach, Germany) and stored at 4 °C before the preparation of single-cell suspensions. The skin tissues of four individuals were merged as one sample, and one sample was prepared for each group.

All procedures applied for this study were approved by the Institutional Ethics Committee of Animal Ethical and Welfare Committee of Chengdu Institute of Biology, Chinese Academy of Sciences (permit: CIBDWLL20220527), and all methods were carried out in accordance with the Code of Practice for the Care and Handling of Animal Guidelines. This study is reported in compliance with the ARRIVE guidelines.

### 2.2. Tissue Dissociation and Preparation of Single-Cell Suspensions

For digestion, 10 mg of collagenase was dissolved in 10 mL of buffer to create a digestion solution, which was sterilized by filtration through a 0.22 µm filter. The minced tissue was transferred to a 50 mL centrifuge tube containing 5 mL of collagenase digestion solution and incubated in a 37 °C water bath with shaking for 15–30 min. After digestion, the solution was filtered through a 40 µm cell strainer to separate single cells. The filtrate was collected in a new sterile centrifuge tube and centrifuged at 1500 rpm for 10 min at 4 °C. The supernatant was discarded, and the cell pellet was resuspended in 10 mL of Ca^2+^- and Mg^2+^-free DPBS containing 2% FBS, followed by centrifugation at 1000 rpm for 5 min. The supernatant was discarded. The cell pellet was resuspended in 3 mL of Ca^2+^- and Mg^2+^-free DPBS (containing 2% FBS) and carefully layered onto 6 mL of a debris removal solution. The suspension was centrifuged at 900 g at room temperature for 20–30 min with slow acceleration and deceleration. The dark cell layer was collected, resuspended in 10 mL of Ca^2+^- and Mg^2+^-free DPBS (containing 2% FBS), and centrifuged at 1000 rpm for 10 min. The supernatant was discarded. The cell pellet was washed with 5 mL of Ca^2+^- and Mg^2+^-free DPBS (containing 2% FBS), centrifuged at 1000 rpm for 5 min at 4 °C, and the supernatant was discarded. The pellet was resuspended in 200 µL of Ca^2+^- and Mg^2+^-free DPBS (containing 2% FBS). Cell counting and viability assays were performed, and appropriate volumes of cell suspension were used for subsequent experiments.

### 2.3. Chromium 10x Genomics Sequencing and Analysis

Library construction followed the method that we described previously [35]. The libraries were sequenced using the Illumina NovaSeq 6000 sequencing system (paired-end multiplexing run, 150 bp) by Personalbio Co., Ltd. (Shanghai, China). Sequencing results were demultiplexed and converted to FASTQ format using Illumina bcl2fastq (v2.20, Illumina, San Diego, CA, USA). Sample demultiplexing, barcode processing, and single-cell 3′ gene counting were performed with the Cell Ranger pipeline (v3.1.0, 10x Genomics, Pleasanton, CA, USA). The scRNA-seq data were aligned to the unpublished *R. dugritei* reference genome (China National GeneBank Nucleotide Sequence Archive with accession number CNP0002375) [36].

Then, we processed the 10x single-cell RNA sequencing data using Seurat (version 4.4.0) [37], creating a Seurat object and setting the parameter min.cells = 1 to retain genes expressed in at least one cell. Quality control criteria were set as nFeature_RNA between 200 and 4000, nCount_RNA between 200 and 40,000, and mitochondrial gene expression (percent.mt) below 20%. After filtering cells that met these criteria, we performed principal component analysis (PCA) and used the ElbowPlot (ndims = 50) to determine the optimal number of principal components. To examine the distribution of the data before batch correction, we applied UMAP (using dims = 1:25, n_neighbors = 50), identifying potential separation or clustering among different sample groups. Next, Harmony (version 1.2.0) [38] was used to correct for batch effects across sample groups, and post-correction, we performed UMAP again (reduction = ‘harmony’, dims = 1:25) to ensure data consistency and reduce technical artifacts.

We visualized the clustering results across varying resolutions using clustree (version 0.5.1) [39], exploring resolution settings from 0 to 0.05 with increments of 0.01 to detect trends in cell population changes. Based on the stability of red blood cells, we selected a final resolution of 0.04 for optimal clustering of cell populations. Subsequently, we used Seurat’s FindAllMarkers function (with parameters test.use = “wilcox”, only.pos = TRUE, min.pct = 0.1, logfc.threshold = 0.25) to identify marker genes for each cluster, exporting the results to CSV files. The gene IDs were obtained based on the genome assembly and annotation information data from the *R. dugritei* genome [36]. Marker genes with an adjusted *p*-value < 0.05 were filtered, and cell types were annotated using the CellMarker2.0 database [40].

The identified cell types and their corresponding marker genes are as follows: Cluster 0 consists of epithelial cells marked by *EPCAM* and *ANXA1*. Cluster 1 contains fibroblasts, characterized by the expression of *FBN1*, *POSTN*, and *MMP2*. Cluster 2 comprises mucus-secreting-like cells, identified by *SPDEF*, *MUC5B*, and *PIGR*. Cluster 3 includes myofibroblasts with the markers *MYLK* and *FSTL1*. Cluster 4 consists of ionocyte-like cells, with a broad range of markers including *CFTR*, *ANK2*, *CNN3*, *COX5*, *ITPR2*, *PYGB*, *CELF2*, *RGMA*, *CGNL1*, *FRYL*, and *AKAP2*. Cluster 5 is composed of red blood cells, identified by *HBB*, *HBAB*, and *RHD*. Cluster 6 includes epithelial cells marked by *EPCAM*, *KRT19 (K1C19)*, *MUC5B*, *OCLN*, *CDH1*, *CLDN3*, and *DSP*. Cluster 7 represents B cells, characterized by *CD22* and *PTPRC (CD45)*. Cluster 8 consists of chromatophores, with marker genes *TYRP1*, *PMEL*, *MITF*, *DCT*, and *GCH1*. Cluster 9 includes epithelial cells identified by *EPCAM* and *FOXM1*. Cluster 10 is made up of basal cells expressing *P63*, *DSC3*, *FAT2*, *GPC1*, *HSPB1*, *TIMP1*, *PKP3*, and *ACKR3*. Cluster 11 consists of smooth muscle cells characterized by *MYH11*, *ACTA*, *ACTC*, and *CNN1*. Cluster 12 includes epithelial cells marked by *KRT19, FOXA1,* and *CDH1*, while Cluster 13 comprises merkel cells expressing *ISL1*, *SOX2*, and *ATOH1*. Lastly, we performed a trajectory analysis using Monocle2 [41] to further explore the differentiation and dynamic changes in chromatophores over time.

### 2.4. Statistical Analyses

Differential analyses of gene expression levels were performed using Mann–Whitney U tests. We performed GO and KEGG enrichment analyses on DEGs using KOBAS 3.0 [42]. The graphs were generated using ggplot2 (version 3.5.0) [43] and GraphPad Prism 7.

## 3. Results

### 3.1. Environment-Induced Skin Color Plasticity in R. dugritei

The skin color of *R. dugritei* changed dramatically in response to environmental background colors within two days (Figure 1). On a white background, individuals exhibited predominantly green dorsal skin with white spots under light conditions. These white spots darkened to brown during the dark phase of the photoperiod. On a black background, the dorsal skin turned deep green or black overall, with no noticeable changes in skin color related to the photoperiod.

### 3.2. Identification of Chromatophores from the Skin

The skin tissues of *R. dugritei* were collected during the dark phase of the photoperiod and sent for single-cell transcriptomic analysis. A total of 6708 cells from the white-background group and 6575 cells from the black-background group were sequenced. The average reads per cell were 39,472 and 39,582 for the white- and black-background groups, respectively. The median number of genes per cell was 1011 for the white-background group and 749 for the black-background group.

The skin cells were clustered into 14 clusters using UMAP, which were identified to be epithelial cells (four cells of the same species expressing different representative genes), fibroblasts, mucus-secreting-like cells, ionocyte-like cells, B cells, chromatophores, red blood cells, basal cells, myofibroblasts, smooth muscle cells, and merkel cells using marker genes reported in the literature (Figure 2a,b). The genes related to pigmentation (e.g., *TYRO*, *TYRP1*, *PMEL*, *GCH1*, and *PAX7*) showed specific transcription in the cell cluster of melanophores (Appendix A). To further ensure the accuracy of chromatophore identification, we performed an enrichment analysis on the function of this cell cluster using its feature genes identified through a UMAP clustering analysis. The results highlighted melanosome formation, melanophore differentiation, and melanogenesis (Figure 2c,d), supporting the cluster’s role in pigmentation. The chromatophore accounted for 1.57% of the total number.

### 3.3. Sub-Classification of the Chromatophore

UMAP dimensionality reduction was performed on the chromatophores. This analysis divided the chromatophores into two sub-clusters (Figure 3a), one of which displayed more robust transcription of protein QNR-71 (*QNR71*) and tyrosinase-related protein 1 (*TYRP1*) (Figure 3b), two critical genes in melanophore differentiation and melanogenesis [44,45]. The other sub-cluster was featured by higher transcription of GTP cyclohydrolase 1 (*GCH1*) and paired box protein pax-7 (*PAX7*) (Figure 3b), which were involved in pterin pigmentation and xanthophore formation [46,47]. Accordingly, these two sub-cluster chromatophores were identified as melanophores and xanthophores, respectively. The expression patterns of pigmentation-related genes were exhibited at the single-cell level (Figure 3c). The results showed that the expression of *TYRP1*, premelanosome protein (*PMEL*), melanoregulin (*MREG*), and *QNR71* was specific to melanophores, while *GCH1* and *PAX7* were specifically expressed in xanthophore.

We analyzed the differentiation trajectory of the chromatophores (Figure 4a). Melanophores and xanthophores did not show any differences in differentiation timing (Figure 4b). A total of 158 genes showed significant transcriptional changes with differentiation pseudotime (Figure 4c), and those upregulated with pseudotime highlighted ribosome, oxidative phosphorylation, and thermogenesis (Figure 4d). These results suggest that mature chromatophores exhibited more robust protein synthesis and energy production.

### 3.4. Cellular and Molecular Basis Underlying Skin Color Differences

In dark-colored frogs, the proportions of melanophores and xanthophores in the skin were significantly higher at the cellular level, with values of 1.78% and 1.25%, respectively, compared to 0.10% and 0.00% in light-colored frogs (Figure 5a,b). We further analyzed the differentially expressed genes (DEGs, *p* < 0.05 for Mann–Whitney U test) in melanophores between light- and dark-colored frogs (Appendix A). Notably, DEGs with higher expression in light-colored individuals were predominantly associated with immune responses, including herpes simplex virus 1 infection and toxoplasmosis pathways (Figure 5c). In contrast, DEGs more highly expressed in dark-colored individuals were primarily related to energy metabolism (Figure 5d). Additionally, we compared the transcriptional levels of core pigmentation genes (Figure 5e). Those positively associated regulating pigmentation (e.g., *TYRO*, *TYRP1*, *PMEL*, *MITF*, *QNR71*, and *DOPDB*) tended to have higher transcription in the dark-colored frogs, but these differences were insignificant (*p* > 0.05, Mann–Whitney U test). In contrast, *MREG*, a gene that suppresses melanization, showed marginally significant downregulation in dark-colored frogs (*p* = 0.054, Mann–Whitney U test) (Figure 5e).

## 4. Discussion

*R. dugritei* frogs displayed remarkable skin color plasticity to the variations in background color and light intensity. Through the scRNA-seq technique, we have successfully identified two primary types of chromatophores, melanophores and xanthophores, in their skin tissues. These two chromatophores exhibited distinctive gene expression patterns, especially those involved in their respective pigmentation synthesis (e.g., *TYRP1*, *PMEL*, *QNR71*, *GCH1*, and *PAX7*). We did not identify any cell clusters suspected to be erythrophores. Genes specific to erythrophores (e.g., *BCO1* and *BCO2*) were identified to be transcribed only in a few cells that were classified as melanophores, and their transcriptional levels were low. Melanophores and xanthophores are derived from neural crest cells, and their differentiation trajectories have been well studied in fish [18,48]. In comparison, the differentiation and physiology of erythrophores remain largely unknown [49]. ScRNA-seq technology presents an opportunity to address these research gaps; however, due to the low proportion of chromatophores in the skin, improved sampling and enrichment methods are needed to target these cells specifically. Currently, skin pigmentation studies at the single-cell level are well established in mammals but remain relatively limited in lower vertebrates [31,32,50,51,52], which exhibit a wider diversity of chromatophore types and a greater capacity to adapt to environmental background colors. Our findings may provide valuable insights for pigmentation research in non-model animals.

Both melanophores and xanthophores play critical roles in background adaptation in *R. dugritei* frogs, with their relative abundance corresponding to the observed skin tone. This suggests that an increase in chromatophore number is a key process contributing to the darker skin color observed under black-background conditions (Figure 6). Additionally, melanophores from dark-colored individuals tended to exhibit higher transcription levels of *TYRO*, *TYRP1*, and *PMEL* compared to those from light-colored individuals, though these differences were not statistically significant. This implies that the melanin synthesis capacity in individual melanophores might also vary with environmental conditions. In addition to MCC, the transcriptional changes in *MREG* implied the existence of PCC in the background adaptation of *R. dugritei* frogs. *MREG* suppresses skin pigmentation by regulating intracellular and intercellular melanosome transfer [53]. The overexpression of *MREG* in normal melanophores caused perinuclear melanosome aggregation, while knockdown of *MREG* restored peripheral melanosome distribution [54]. In addition, the PCC mechanism is equally influenced by environmental factors (such as light and temperature) and hormonal factors (such as progesterone and testosterone, α-MSH, and β-epinephrine), which directly affect the behavior of melanosomes [2,55,56]. In vitro studies on melanocytes of other lower vertebrates have shown that these factors can directly regulate the dispersion and aggregation of melanosomes, suggesting that the PCC process can be regulated by multiple factors [55,57,58]. PCC was likely secondary to MCC in determining the relationship between skin and background color in *R. dugritei* frogs. Nevertheless, PCC, which is affected by multiple factors, is still worthy of in-depth study, and a broader range of techniques, such as metabolomics, are needed to study the interactions between these factors and chromatophores. Previous studies on background matching mainly focused on color changes at time scales of minutes and hours and thus considered PCC as a major mechanism of background color matching [24,25,26,28]. In fact, seasonal color changes in animals hold significance in their environmental adaptation [59]. The role of MCC, as well as its interactions with other physiological activities, in background adaptation should receive more attention.

Our data also reveal significant molecular differences in melanophores between light- and dark-colored frogs. Beyond the variation in pigmentation-related genes, melanophores from dark-colored individuals exhibited higher transcriptional activity in energy metabolism pathways. A pseudotime analysis of chromatophore differentiation showed that mature chromatophores had enhanced potential for protein synthesis and energy production, as evidenced by the upregulation of genes associated with ribosomal activity, oxidative phosphorylation, and thermogenesis. Together, these findings imply that melanophores in darker-colored frogs are more mature compared to those in light-colored individuals (Figure 6). Meanwhile, melanophores in light-colored frogs exhibited stronger transcription of genes related to immune responses. In vertebrates, melanophores can regulate skin immune responses by producing various cytokines, such as IL-1, IL-6, IL-3, and TNFα [60], with infection and inflammatory responses influencing both the immune and metabolic functions of melanophores [61], this mechanism may arise because light-colored animals are more susceptible to UV damage [62]. Thus, background colors may not only affect the morphology of animals but also impact their immune responses.

In summary, our findings illuminate the cellular and molecular mechanisms underlying skin color adaptation in *R. dugritei*. MCC driven by melanophore and xanthophore proliferation, rather than MCC and PCC of individual cells, is the primary factor contributing to their background adaptation. Additionally, the transcriptional changes observed in chromatophores across different color states suggest a strong link between pigmentation, energy metabolism, and immune responses—a relationship that has been noted in previous studies.

## 5. Conclusions

Our study provides new insights into the cellular and molecular mechanisms underlying rapid skin color changes in *R. dugritei*. We demonstrate that melanophores and xanthophores play pivotal roles in this adaptive response, with both physiological and morphological color changes working synergistically to drive the process. This finding not only advances our understanding of the mechanisms underlying color adaptation in amphibians but also offers valuable insights into adaptive color changes in other non-model vertebrates. Future investigations will focus on elucidating the long-term roles of these mechanisms in facilitating animal adaptation to environmental changes. Furthermore, these findings hold significant implications for the conservation of endangered species and offer promising avenues for biomimetic applications, expanding our understanding of and potential for leveraging biological adaptability in biomedical and technological innovations.

## Figures and Tables

**Figure 1 animals-14-03472-f001:**
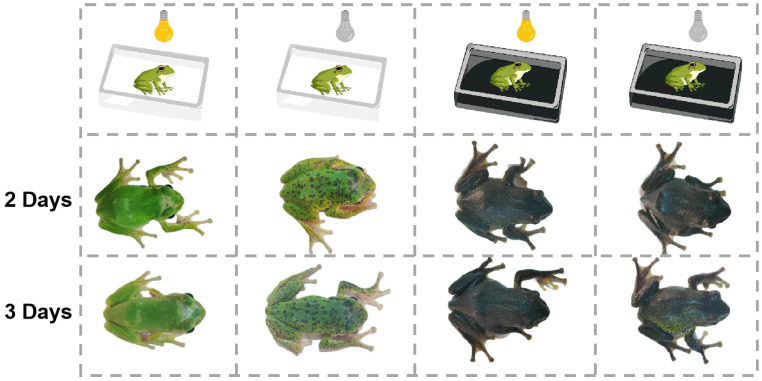
The morphology of *R. dugritei* exposed to different environments. The skin color of this animal varied with both background color and photoperiod (light phase–dark phase = 12:12).

**Figure 2 animals-14-03472-f002:**
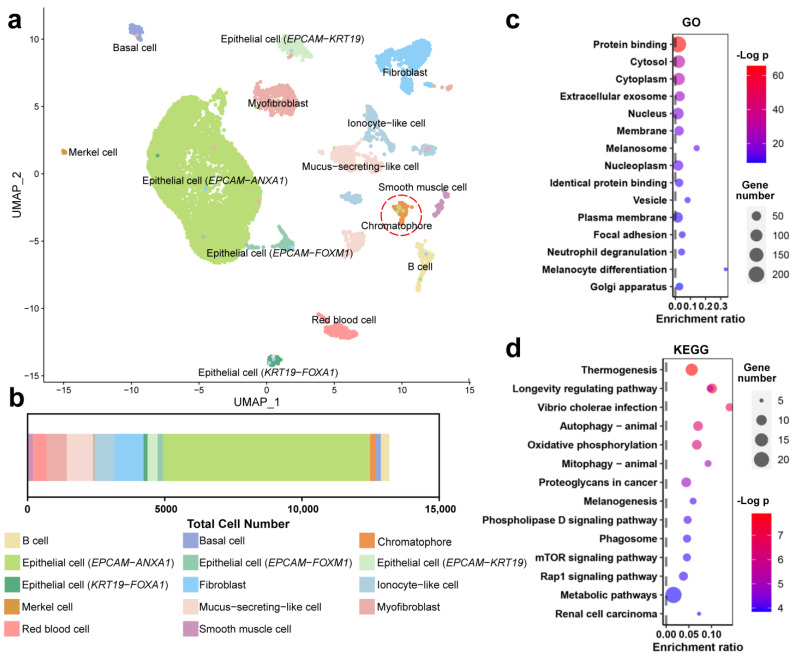
Single-cell analysis of skin tissue. (**a**) Cell type annotation. The cell cluster standing for chromatophores is highlighted by a red dashed circle. (**b**) Cell proportions of the skin tissue of *R. dugritei*. (**c**,**d**) KEGG enrichment analyses of the feature genes of chromatophores. The pathways related to pigmentation are highlighted by red dashed circles.

**Figure 3 animals-14-03472-f003:**
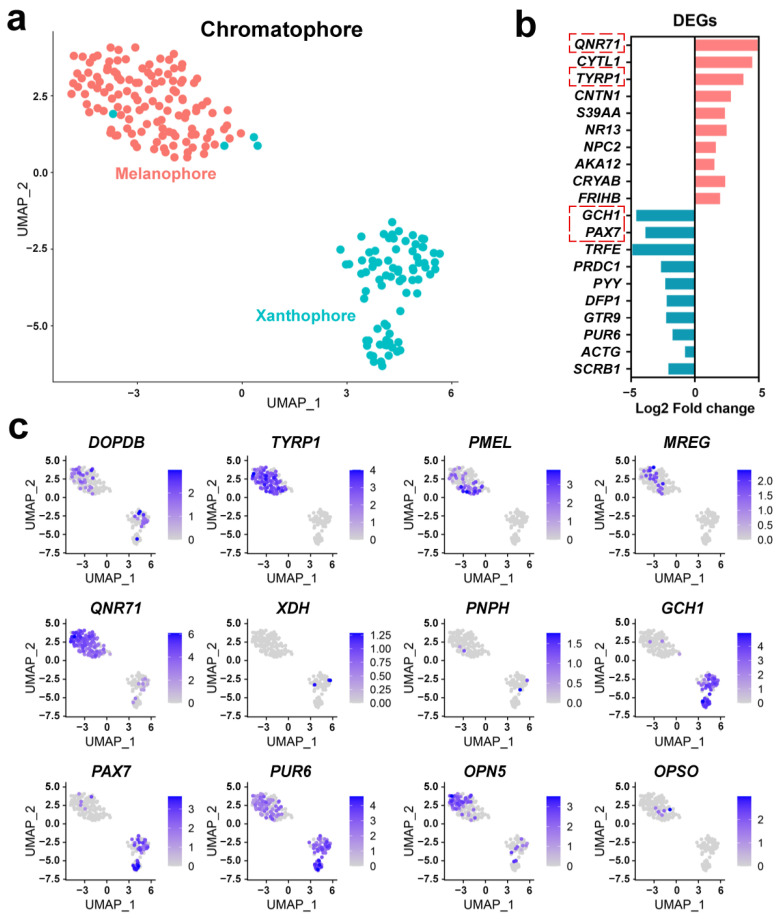
Classification and subtype identification of chromatophores. (**a**) Further separation of the chromatophores into two sub-clusters in UMAP. (**b**) Featured genes of each sub-cluster. The fold changes were calculated as the ratio of the average value in melanophores to the average value in xanthophores. The critical genes involved in pigmentation are highlighted by red dashed boxes. (**c**) Relative expression of pigmentation-related genes in each chromatophore. Gene abbreviations: *ACTG*, actin (cytoplasmic 2); *AKA12*, A-kinase anchor protein 12; *CNTN1*, contactin 1; *CRYAB*, alpha-crystallin b chain; *CYTL1*, cytokine-like protein 1; *DFP1*, putative defense protein 1; *DOPDB*, D-dopachrome decarboxylase-B; *FRIHB*, ferritin heavy chain; *GCH1*, GTP cyclohydrolase 1; *GTR9*, solute carrier family 2 (facilitated glucose transporter member 9); *MREG*, melanoregulin; *OPN5*, opsin 5; *NPC2*, NPC intracellular cholesterol transporter 2; *NR13*, anti-apoptotic protein NR13; *OPSO*, opsin family member; *PAX7*, paired box protein pax-7; *PMEL*, premelanosome protein; *PNPH*, purine nucleoside phosphorylase; *PRDC1*, phosphoribosyltransferase domain-containing protein 1; *PUR6*, multifunctional protein ADE2; *PYY*, peptide YY; *QNR71*, protein QNR-71; *S39AA*, zinc transporter ZIP10; *SCRB1*, scavenger receptor class b member 1; *TYRP1*, tyrosinase-related protein 1; *XDH*, xanthine dehydrogenase; *TRFE*, transferrin.

**Figure 4 animals-14-03472-f004:**
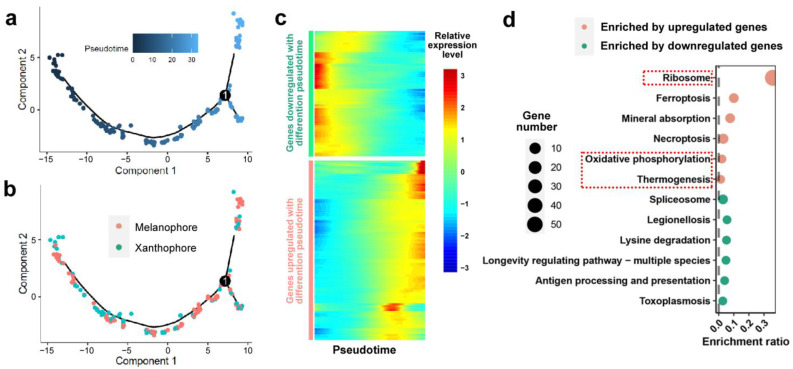
Trajectory analysis of chromatophores. (**a**) Pseudotime differentiation trajectory of chromatophores. The lighter the color, the lower the degree of differentiation and thus the less mature in function. (**b**) The distribution of the two types of chromatophores in the pseudotime differentiation trajectory. (**c**) Heat maps illustrating the transcriptional variation in differentially expressed genes (DEGs) in relation to differentiation pseudotime. Each row represents a gene, while the columns correspond to chromatophores arranged from left to right according to differentiation pseudotime, from low to high. (**d**) KEGG enrichment analyses based on DEGs associated with differentiation pseudotimes. Mature chromatophores are featured with robust transcription of ribosome and energy metabolism (highlighted by red dashed boxes).

**Figure 5 animals-14-03472-f005:**
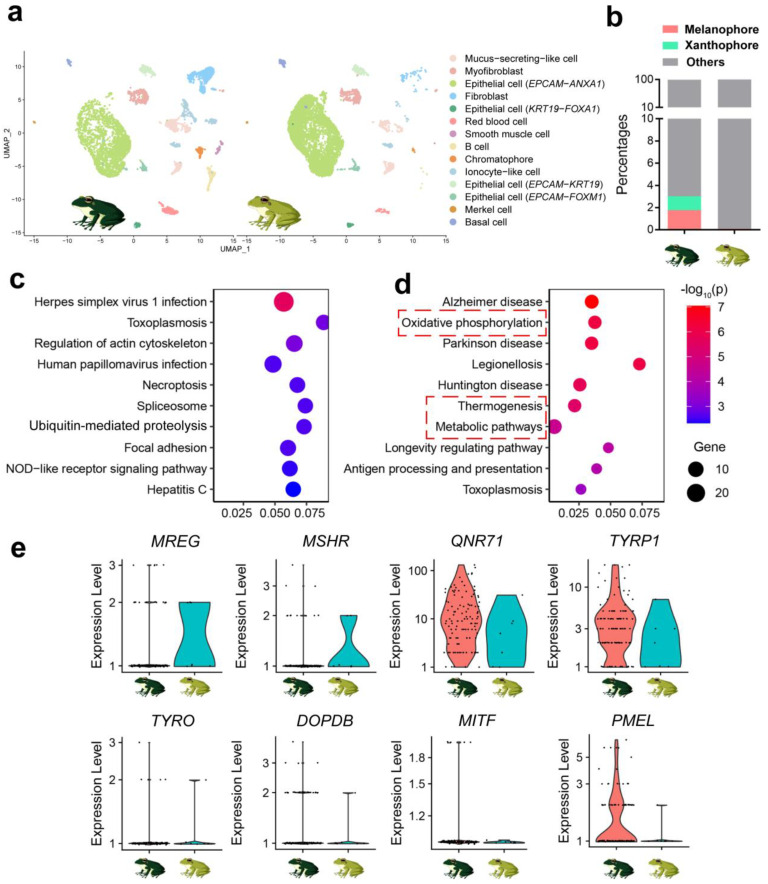
Cellular and molecular differences in skin tissue between individuals from white and black backgrounds. (**a**) UMAP scatter plots presenting the differences in cell composition between groups. The chromatophores are highlighted with red dashed circles. (**b**) Proportions of chromatophores. (**c**,**d**) Results of KEGG enrichment analyses based on DEGs showing lower (**c**) and higher (**d**) expression levels (*p* < 0.05) in melanophores of the black-background group. The top 10 most significantly (adjusted *p* < 0.05) enriched KEGG items are displayed. The KEGG items related to energy metabolism are highlighted with red dashed boxes. (**e**) Transcriptional levels of core genes involved in pigmentation in melanophores. Gene abbreviations: *DOPDB*, D-dopachrome decarboxylase-B; *MITF*, microphthalmia-associated transcription factor; *MREG*, melanoregulin; *MSHR*, melanocortin 1 receptor; *PMEL*, premelanosome protein; *QNR71*, protein QNR-71; *TYRO*, tyrosinase; *TYRP1*, tyrosinase-related protein 1.

**Figure 6 animals-14-03472-f006:**
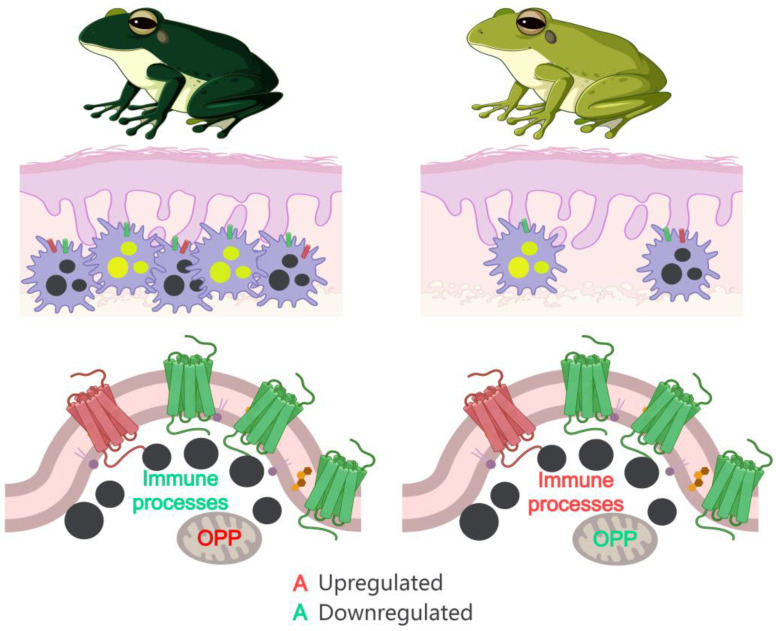
The cellular and molecular basis for the environment-induced color change in *R. dugritei* frogs. The color of the text represents the gene or process’s association with pigmentation, while the size of the text indicates its relative transcription level. OPP represents Oxidative Phosphorylation.

## Data Availability

All data needed to evaluate the conclusions in the paper are available in the GSA (CRA019688) or present in the paper and/or the Appendix A. Additional data related to this paper may be requested from the authors.

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
