# Peer review of "Cellular and Molecular Basis of Environment-Induced Color Change in a Tree Frog"

_animals, 2024, doi:10.3390/ani14233472_

Round 1
Reviewer 1 Report
Comments and Suggestions for Authors
The study investigates the cellular and molecular mechanisms underlying the remarkable ability of the tree frog species Rhacophorus dugritei to change its skin color in response to different environmental backgrounds (white or black). Using single-cell RNA sequencing (scRNA-seq) of skin samples, the authors identified two distinct types of pigment cells (chromatophores) – melanocytes and xanthophores – responsible for the observed color changes. The findings suggest that morphological color change (MCC), driven by changes in the number of chromatophores rather than their individual activity, is the dominant process in this species' background adaptation. Additionally, the authors observed increased energy metabolism in melanocytes from darker frogs and heightened immune-related gene expression in lighter-colored individuals, indicating that background adaptation involves more than just pigmentation changes.
This is one of the first studies to apply single-cell sequencing technology to investigate skin pigmentation in a non-model organism, providing insights into the cellular mechanisms of background adaptation in amphibians. The study addresses a long-standing question in the field – whether the number of chromatophores or their individual activity plays a more critical role in vertebrate color change. The identification of two distinct chromatophore types (melanocytes and xanthophores) involved in background adaptation, and the discovery that MCC driven by changes in chromatophore numbers is the dominant process, offer new insights into amphibian color adaptation mechanisms.
After a thorough review of the manuscript titled 'Cellular and Molecular Basis of Environment-Induced Color Change in a Tree Frog,' I would recommend accepting this paper for publication, provided that one minor concern is addressed. Specifically, additional details regarding the quality and completeness of the unpublished reference genome used in this study would be valuable. Such information is crucial for assessing the accuracy of the bioinformatics analyses presented.
Author Response
Comments 1: The study investigates the cellular and molecular mechanisms underlying the remarkable ability of the tree frog species Rhacophorus dugritei to change its skin color in response to different environmental backgrounds (white or black). Using single-cell RNA sequencing (scRNA-seq) of skin samples, the authors identified two distinct types of pigment cells (chromatophores) – melanocytes and xanthophores – responsible for the observed color changes. The findings suggest that morphological color change (MCC), driven by changes in the number of chromatophores rather than their individual activity, is the dominant process in this species' background adaptation. Additionally, the authors observed increased energy metabolism in melanocytes from darker frogs and heightened immune-related gene expression in lighter-colored individuals, indicating that background adaptation involves more than just pigmentation changes.
This is one of the first studies to apply single-cell sequencing technology to investigate skin pigmentation in a non-model organism, providing insights into the cellular mechanisms of background adaptation in amphibians. The study addresses a long-standing question in the field-whether the number of chromatophores or their individual activity plays a more critical role in vertebrate color change. The identification of two distinct chromatophore types (melanocytes and xanthophores) involved in background adaptation, and the discovery that MCC driven by changes in chromatophore numbers is the dominant process, offer new insights into amphibian color adaptation mechanisms.
After a thorough review of the manuscript titled 'Cellular and Molecular Basis of Environment-Induced Color Change in a Tree Frog,' I would recommend accepting this paper for publication, provided that one minor concern is addressed. Specifically, additional details regarding the quality and completeness of the unpublished reference genome used in this study would be valuable. Such information is crucial for assessing the accuracy of the bioinformatics analyses presented.
Response 1:
We sincerely thank you for your valuable feedback and constructive suggestion regarding the details of the reference genome used in our study. We fully agree that the quality and completeness of the genome are crucial for assessing the reliability of bioinformatics analyses.
The R. dugritei genome used in our study is derived from a high-quality, previously published genome (reference [36]) (Line 165). This genome was assembled using a hybrid approach that integrated data from multiple sequencing platforms, including 10× Genomics Chromium, PacBio Sequel, and Hi-C sequencing on the Illumina NovaSeq 6000 platform. The assembly strategy employed a combination of de novo methods, ab initio, and homology-based annotations, resulting in a comprehensive and highly accurate reference genome. Additionally, the supplementary materials of the published study (SI Appendix, Tables S20–S22) offer extensive documentation of the sequencing, resequencing, and annotation processes, ensuring transparency and verifiability of the data.
We have also included content in the acknowledgments to express our gratitude to the original authors for publishing this data. We appreciate your insightful comments, which have allowed us to highlight this crucial aspect of our methodology. Thank you for recognizing the potential of our research and for recommending it for publication.
Reviewer 2 Report
Comments and Suggestions for Authors
“evolution, and conservation..” extra dot added
“In ectothermic vertebrates, the main types of pigment cells (or called chromatophores) include melanophores, xanthophores, and erythrophores” However, In ectothermic vertebrates, there are indeed multiple types of chromatophores, each contributing to the diverse coloration seen in these animals. The primary types include:
- Melanophores: These contain melanin, giving a black or brown coloration.
- Xanthophores: These contain yellow pigments, such as pteridines or carotenoids.
- Erythrophores: These contain red pigments, typically carotenoids.
- Iridophores (or Guanophores): These contain reflective, iridescent platelets of guanine, creating a shiny or metallic appearance.
- Leucophores: These contain white, non-pigmented components and reflect light to create a white or pale appearance.
- Cyanophores (found in some species, such as certain fish): These contain blue pigments.
“and and whether certain” The word and is written twice
“environments. 2. Materials and Methods” Materials and methods title is combined with the sentence above.
Authors should clarify that why they use only male samples.
at 36 (L), 48 (D), 60 (L), and 72 (D) authors should clarify L and D.
“1 × 1 cm2” 2 should be written as superscript
1,500 rpm no need comma
1,000 rpm no need comma
1,000 rpm no need comma
6,708 cells no need comma
6,575 cells from no need comma
39,472 and 39,582 no need comma
1,011 for no need comma
The discussion section effectively presents the findings on skin color adaptation in R. dugritei and integrates relevant literature, showcasing the roles of melanocytes and xanthophores. However, it lacks a strong conclusion and could better emphasize the novel contributions of the study. While it touches on pigmentation, immune responses, and energy metabolism, the section would benefit from deeper mechanistic insights and a clearer articulation of how these elements interact. Finally, adding specific suggestions for future research would strengthen the section's relevance and encourage continued investigation into pigment cell adaptation in non-model species.
To improve the conclusion, authors could:
Emphasize Novelty: Mention that your study provides unprecedented insights into rapid skin color changes at the cellular level.
Broader Implications: Note the potential relevance for adaptive coloration in other vertebrates.
Future Directions: Suggest further exploration of long-term environmental impacts on these mechanisms.
Impact Statement: Highlight the broader significance for conservation and biomimetics.
Author Response
Comments 1: “evolution, and conservation..” extra dot added.
Response 1: Thank you for the reminder; we have removed the extraneous period (Line 19).
Comments 2: “In ectothermic vertebrates, the main types of pigment cells (or called chromatophores) include melanophores, xanthophores, and erythrophores”. However, in ectothermic vertebrates, there are indeed multiple types of chromatophores, each contributing to the diverse coloration seen in these animals. The primary types include:
1. Melanophores: These contain melanin, giving a black or brown coloration.
2. Xanthophores: These contain yellow pigments, such as pteridines or carotenoids.
3. Erythrophores: These contain red pigments, typically carotenoids.
4. Iridophores (or Guanophores): These contain reflective, iridescent platelets of guanine, creating a shiny or metallic appearance.
5. Leucophores: These contain white, non-pigmented components and reflect light to create a white or pale appearance.
6. Cyanophores (found in some species, such as certain fish): These contain blue pigments.
Response 2:
We sincerely thanks for your insightful feedback and for emphasizing the diversity of chromatophore types in ectothermic vertebrates. We fully agree that a more detailed description of these pigment cells would greatly enhance the reader’s understanding of their roles in coloration.
In response to this comment, the revised introduction now includes a description of the six types of chromatophores—melanophores, xanthophores, erythrophores, iridophores, leucophores, and cyanophores(Lines 48-56). We have highlighted their unique pigment compositions and how they contribute to the diverse coloration seen in ectothermic vertebrates. This expanded section aims to provide readers with a thorough understanding of the biological diversity and functions of these chromatophores.
We are grateful for your constructive suggestion, which has significantly enhanced the depth and clarity of our manuscript. Thank you once again for your valuable feedback and continued support.
Lines 48-56: “Chromatophores are pigment cells in ectothermic vertebrates that generate color by producing pigments or reflecting light. They facilitate dynamic color changes, allowing adaptation to environmental conditions. Chromatophores are classified into six types: melanophores (black or brown melanin), xanthophores (yellow pigments from pteridines and carotenoids), erythrophores (red pigments from carotenoids); iridophores and leucophores (guanine platelets for metallic or reflective white colors); and cyanophores (blue pigmentation in certain species, such as fish. The interplay and regulation of these types, driven by specific pigment synthesis genes, underpin the complex color changes observed in ectothermic vertebrates.”
Comments 3: Authors should clarify that why they use only male samples.
Response 3:
Thank you for raising this important point regarding the use of only male samples in our study. In response, we have added a clarification in the experimental design section of the manuscript (Lines111-114). Specifically, we selected only male frogs to minimize potential variability in hormone levels and gene expression that could arise from sex differences. Besides, the proportion of females in this species is low in the field, and collecting female individuals could have a substantial impact on the population. This approach was employed to ensure consistency and reliability in our subsequent analyses. Thanks for your valuable comments.
Lines 111-114: “Only male individuals were included in this study to minimize the effects of sex differences on hormone levels and gene expression. Additionally, this species had a low proportion of females in the wild, the collection of female individuals could have a significant impact on the population.”
Comments 4:
“and and whether certain” The word and is written twice
“environments. 2. Materials and Methods” Materials and methods title is combined with the sentence above.
at 36 (L), 48 (D), 60 (L), and 72 (D) authors should clarify L and D.
“1 × 1 cm2” 2 should be written as superscript
1,500 rpm no need comma
1,000 rpm no need comma
1,000 rpm no need comma
6,708 cells no need comma
6,575 cells from no need comma
39,472 and 39,582 no need comma
1,011 for no need comma
Response 4:
Thanks for this meticulous review and for pointing out these formatting and clarity issues. We have carefully addressed all the mentioned points in the revised manuscript, as follows:
- Removed the repeated word "and" in the text (Line 80).
- Corrected the formatting issue where the "Materials and Methods" title was combined with the sentence above (Line 108).
- Clarified the meaning of "L" and "D" in the relevant section (Lines 118, 125-126, and 219).
- Adjusted "1 × 1 cm2" to ensure the "2" is correctly written as a superscript (Line 128)
- Removed unnecessary commas in the numerical values (e.g., 1,500 rpm → 1500 rpm, 6,708 cells → 6708 cells, etc.).
Comments 5: The discussion section effectively presents the findings on skin color adaptation in R. dugritei and integrates relevant literature, showcasing the roles of melanocytes and xanthophores. However, it lacks a strong conclusion and could better emphasize the novel contributions of the study. While it touches on pigmentation, immune responses, and energy metabolism, the section would benefit from deeper mechanistic insights and a clearer articulation of how these elements interact. Finally, adding specific suggestions for future research would strengthen the section's relevance and encourage continued investigation into pigment cell adaptation in non-model species.
To improve the conclusion, authors could:
1. Emphasize Novelty: Mention that your study provides unprecedented insights into rapid skin color changes at the cellular level.
2. Broader Implications: Note the potential relevance for adaptive coloration in other vertebrates.
3. Future Directions: Suggest further exploration of long-term environmental impacts on these mechanisms.
4. Impact Statement: Highlight the broader significance for conservation and biomimetics.
Response 5: Thanks for your thoughtful feedback and suggestions to strengthen the discussion and conclusion sections of our manuscript. In response, we have revised the Conclusions section to better emphasize the novelty of our findings, their broader implications, and future research directions. The updated Conclusions now highlight the following points:
1. Emphasis on Novelty: We explicitly state that our study provides unprecedented insights into the cellular and molecular mechanisms underlying rapid skin color adaptation in R. dugritei. (Lines 394-395: “Our study provides new insights into the cellular and molecular mechanisms underlying rapid skin color changes in R. dugritei.”)
2. Broader Implications: We discuss how these findings advance the understanding of adaptive coloration across vertebrates, suggesting potential relevance beyond amphibians. (Lines 395-400: “We demonstrate that melanophores and xanthophores play pivotal roles in this adaptive response, with both physiological and morphological color changes working synergistically to drive the process. This finding not only advances our understanding of the mechanisms underlying color adaptation in amphibians but also offers valuable insights into adaptive color changes in other non-model vertebrates.”)
3. Future Directions: We propose further exploration of the long-term roles of these mechanisms in environmental adaptation, including their impact on ecosystem dynamics and co-adaptive evolution. (Lines 400-401: “Future investigations will focus on elucidating the long-term roles of these mechanisms in facilitating animal adaptation to environmental changes”)
4. Impact Statement: We underscore the significance of our findings for conservation biology and biomimetics, suggesting promising applications in biomedical and technological innovations. (Lines 401-405: “Furthermore, these findings hold significant implications for the conservation of endangered species and offer promising avenues for biomimetic applications, expanding our understanding of and potential for leveraging biological adaptability in biomedical and technological innovations.”)
We believe these revisions align well with your suggestions and enhance the overall relevance and impact of the manuscript. Thank you again for your constructive feedback and support.
Reviewer 3 Report
Comments and Suggestions for Authors
Please see attached document. Thank you.

The English is appropriate but there are some grammatical errors that must be addressed.
Author Response
General comments:
Comments 1: Please provide clear biological descriptions of chromatophore, xanthophore and melanocyte in the introduction.
Response 1: We sincerely thank you for your valuable suggestion. In response, we have revised the introduction section to improve the description of chromatophores and their subtypes (Lines 48-56), making it clearer.
These revisions enhanced the clarity and comprehensiveness of the introduction. Thank you for your constructive suggestions.
Lines 48-56: “Chromatophores are pigment cells in ectothermic vertebrates that generate color by producing pigments or reflecting light. They facilitate dynamic color changes, allowing adaptation to environmental conditions. Chromatophores are classified into six types: melanophores (black or brown melanin), xanthophores (yellow pigments from pteridines and carotenoids), erythrophores (red pigments from carotenoids); iridophores and leucophores (guanine platelets for metallic or reflective white colors); and cyanophores (blue pigmentation in certain species, such as fish. The interplay and regulation of these types, driven by specific pigment synthesis genes, underpin the complex color changes observed in ectothermic vertebrates.”
Comments 2: In the discussion, the results of the study should be compared to some in vitro experiments (there are several on pigment production in frogs)e.g.https://onlinelibrary.wiley.com/doi/abs/10.1111/j.1600-0749.1990.tb00260.x
Response 2: Thanks for your insightful suggestion to compare our findings with results from in vitro experiments. Based on relevant studies, we have added the following description at Lines 352-355: “In vitro studies on melanocytes of other lower vertebrates have shown that these factors can directly regulate the dispersion and aggregation of melanosomes, suggesting that the PCC process can be regulated by multiple factors [56, 58, 59]”, we have expanded the discussion section to include a comparative analysis of PCC mechanisms.
References:
56, Himes PJ, Hadley ME: In vitro effects of steroid hormones on frog melanophores. The Journal of investigative dermatology 1971, 57 5:337-342.
58, Daniolos A, Lerner AB, Lerner MR: Action of light on frog pigment cells in culture. Pigment cell research 1990, 3 1:38-43.
59, Wakamatsu Y, Kawamura S, Yoshizawa T: Light-induced pigment aggregation in cultured fish melanophores: spectral sensitivity and inhibitory effects of theophylline and cyclic adenosine-3′,5′-monophosphate. Journal of Cell Science 1980, 41(1):65-74.
Thank for for bringing this connection to our attention, as it has allowed us to enrich the discussion and contextualize our findings within the broader field of pigment cell research. Thank you for your valuable feedback, which has greatly enhanced the manuscript.
Comments 3: What type of lighting was used for the experiments? What intensity? What kind of bulb?
Response 3: Thank you for raising this important point regarding the lighting conditions used in our experiments. In response, we have revised the manuscript to provide detailed information on the type, intensity, and source of lighting (Lines 116-120).
Lines 116-120: “These animals were acclimated to an artificial climate incubator (YANGHUI Instrument Co., Ltd., Ningbo, China) for seven days at 20 ± 1 ℃ with a 12-hour light/12-hour dark cycle. Illumination was provided by the incubator’s built-in LED lamps, set to 20% of the maximum output (equivalent to approximately 4000 Lx, based on a range of 0-20000 Lx).”
Thank you for your thoughtful suggestion.
Comments 4: Since only males were used in the study, what about the effects of hormones on melanin production? Testosterone causes a dispersion in melanophores in frogs. https://onlinelibrary.wiley.com/doi/10.1111/pcmr.12040
Response 4: Thank for your important questions regarding the potential effects of hormones, such as testosterone, on melanin production and melanophore dispersion. It’s a pit that we have not specifically focused on the influence of sex hormones on pigmentation. But we added a discussion of hormonal factors to the revision (Lines 349-352). Upon revisiting our data, we failed to identify the testosterone receptor in the pigment cells. This may be due to the fact that single-cell sequencing technology can identify only a limited number of genes per cell. Additionally, as we focused on the transcriptional variations of skin (rather than the glands), we were unable to obtain data reflecting testosterone levels. However, your suggestion has offered a valuable direction for future work. Moving forward, we plan to employ a broader range of techniques (e.g., metabolomics) to investigate the interactions between the environment, hormones, and pigment cells. We have incorporated this idea into our discussion section (Lines 357-360). We hope our responses can address your concern. Thanks again for your valuable comments.
Lines 349-352: “In addition, the PCC mechanism is equally influenced by environmental factors (such as light and temperature) and hormonal factors (such as progesterone and testosterone, α-MSH and β-epinephrine), which directly affect the behavior of melanosomes[2, 56, 57]”
Lines 357-360: “PCC was likely secondary to MCC in determining the relationship between skin and background color in R. dugritei frogs. Nevertheless, PCC, which is affected by multiple factors, is still worthy of in-depth study, and a broader range of techniques, such as metabolomics, are needed to study the interactions between these factors and chromatophores.”
References:
2, Nilsson Sköld H, Aspengren S, Wallin M: Rapid color change in fish and amphibians–function, regulation, and emerging applications. Pigment cell & melanoma research 2013, 26(1):29-38.
56, Himes PJ, Hadley ME: In vitro effects of steroid hormones on frog melanophores. The Journal of investigative dermatology 1971, 57 5:337-342.
57, Roubos EW, Van Wijk DCWA, Kozicz T, Scheenen WJJM, Jenks BG: Plasticity of melanotrope cell regulations in Xenopus laevis. European Journal of Neuroscience 2010, 32(12):2082-2086.
Comments 5: Perhaps lighter-colored animals showed more immune gene expression due to UV damage?
Response 5: Thanks for your enlightening suggestions. We have revised the discussion section to incorporate this perspective into the discussion (378-383). Specifically, we now propose that the observed differences in immune gene expression may result from the susceptibility of light-colored animals to UV damage. This addition strengthens the link between background color adaptation and immune responses, providing a plausible explanation for our findings. Thanks for your valuable comments.
Lines 378-383: “Meanwhile, melanophores in light-colored frogs exhibited stronger transcription of genes related to immune responses. In vertebrates, melanophores can regulate skin immune responses by producing various cytokines, such as IL-1, IL-6, IL-3, and TNFα, with infection and inflammatory responses influencing both the immune and metabolic functions of melanophores, this mechanism may arise because light-colored animals are more susceptible to UV damage.”
Specific comments:
Comments 6: 97: Please do not use personal pronouns and change "are" to “were”
Response 6: Thank you for pointing out this for us. We have revised this following your suggestion (Line 100).
Line 100: “The primary scientific questions for this study were…”
Comments 7: 103-104: Please correct “Materials..."
Response 7: Thanks for reminding us of this. We have corrected the formatting (Line 108).
Line 108: “2. Materials and Methods”
Comments 8: 116: What method was used for euthanasia? Please add.
Response 8: Thanks for this important point. The revised text specifies that 0.2% MS-222 was used for euthanasia, ensuring the humane treatment of the animals in accordance with ethical standards (Lines 126-128: “After 72 hours of treatment, the eight frogs were euthanatized using 0.2% MS-222 and collected the dorsal skin tissues (approximately 1 × 1 cm2 per sample).”).
Comments 9: 176-192: Why are some of the genes in bold text?
Response 9: The intention was to clarify in the manuscript that the genes shown in bold represent four type-specific marker genes for epithelial cells. The font style of the gene names has now been standardized to non-bold (Lines 188-203). Thanks for your comment.
Comments 10: 216-219: The authors should remove capital letters from the types of cells.
Response 10: Done. Thanks for your advice. Please see the revisions in Lines 227-231.
Lines 227-231: “The skin cells were clustered into 14 clusters using UMAP, which were identified to be epithelial cell (four cells of the same species expressing different representative genes), fibroblast, mucus-secreting-like cell, ionocyte-like cell, B cell, chromatophore, red blood cell, basal cell, myofibroblast, smooth muscle cell, and merkel cell using marker genes reported in the literature (Figure 2a–b).”
Comments 11: 351-352: In regards to protein synthesis - gene expression does not necessarily correlate with higher levels.
Response 11: Yes. We agree with your point. In the revised version, we emphasized the potential for protein synthesis, as transcription is indeed a major approach for the regulation of protein synthesis. Please see the revisions at Lines 373-376.
Lines 373-376: “Pseudotime analysis of chromatophore differentiation showed that mature chromatophores had enhanced potential for protein synthesis and energy production, as evidenced by the upregulation of genes associated with ribosomal activity, oxidative phosphorylation, and thermogenesis.”
Comments 12: 362 I would disagree with this since chromatophore proliferation was not directly assessed. There is not enough evidence to support this claim. The chromatophores make up a very small percentage of the skin (Figure 2B). However, melanocyte or xanthophore proliferation can be supported due to the observation of PAX7 gene expression.
Response 12:Thanks for your important comments. We have revised the text to replace "chromatophore" with " melanophore and xanthophore," (Line 387) which more accurately reflects the data presented in the manuscript.
We have reviewed the manuscript, corrected grammatical errors, and ensured clarity and precision throughout. Thank you once again for your valuable suggestions.